# High-dose intravenous immunoglobulins might modulate inflammation in COVID-19 patients

María Luisa Rodríguez de la Concepción[1,*] ⓘ, Erola Ainsua-Enrich[1,*] ⓘ, Esteban Reynaga[2,*] ⓘ, Carlos Ávila-Nieto[1], Jose Ramón Santos[2], Silvia Roure[2], Lourdes Mateu[2,4,5], Roger Paredes[1,2], Jordi Puig[2], Juan Manuel Jimenez[2], Nuria Izquierdo-Useros[1] ⓘ, Bonaventura Clotet[1,2,3,4,5], María Luisa Pedro-Botet[2,4,5] ⓘ, Jorge Carrillo[1] ⓘ

The use of high-dose of intravenous immunoglobulins (IVIGs) as immunomodulators for the treatment of COVID-19–affected individuals has shown promising results. IVIG reduced inflammation in these patients, who progressively restored respiratory function. However, little is known about how they may modulate immune responses in COVID-19 individuals. Here, we have analyzed the levels of 41 inflammatory biomarkers in plasma samples obtained at day 0 (pretreatment initiation), 3, 7, and 14 from five hospitalized COVID-19 patients treated with a 5-d course of 400 mg/kg/d of IVIG. The plasmatic levels of several cytokines (Tumor Necrosis Factor, IL-10, IL-5, and IL-7), chemokines (macrophage inflammatory protein-1α), growth/tissue repairing factors (hepatic growth factor), complement activation (C5a), and intestinal damage such as Fatty acid–binding protein 2 and LPS-binding protein showed a progressive decreasing trend during the next 2 wk after treatment initiation. This trend was not observed in IVIG-untreated COVID-19 patients. Thus, the administration of high-dose IVIG to hospitalized COVID-19 patients may improve their clinical evolution by modulating their hyperinflammatory and immunosuppressive status.

## Introduction

Coronavirus infectious disease 2019 (COVID-19) is a new illness caused by the severe acute respiratory syndrome coronavirus 2 (SARS-CoV-2), a novel coronavirus strain that was identified in China at the beginning of 2020 (Wu et al, 2020; Zhu et al, 2020). At present, more than 175 million of infections and 3.8 million of deaths have been reported globally (World Health Organization Coronavirus Disease [COVID-19] Dashboard; https://covid19.who.int/). Although the majority of infected individuals develop mild symptoms, about 10–30% can develop a severe disease that requires hospitalization and, eventually, the admission into the intensive care unit (China CDC Weekly, 2020; European Center for Disease Prevention and Control, 2020). Severe COVID-19 is characterized by pneumonia, dyspnea, hypoxemia, hyperinflammation, and lymphopenia. It can rapidly progress to respiratory failure and acute respiratory distress syndrome, which is associated with a higher risk of death (Yang et al, 2020). An early and sustained increase in cytokine and chemokine levels has been associated with severe cases of the disease (Huang et al, 2020; Lucas et al, 2020). Furthermore, the myeloid cell compartment is also highly deregulated in those patients, in which a high frequency of dysfunctional immature neutrophils with an immunosuppressive profile and a reduction of the nonclassical monocytes has been reported (Schulte-Schrepping et al, 2020; Silvin et al, 2020). Thus, it has been postulated that the hyperinflammatory response induced by SARS-CoV-2 does not only induce a dysregulated response of innate immune cells but may also be behind the tissue damage observed in severe COVID-19 patients (Merad & Martin, 2020). Autopsy studies have shown a strong immune infiltration into the lung in deceased severe COVID-19 individuals that can be categorized in two states of disease progression. The first one is characterized by an accumulation of innate cells associated with a high expression of interferon-related genes (Nienhold et al, 2020). In the second one, tissue infiltrate composed by CD8[+] T cells and macrophages was associated with a lower expression of interferon-related genes, local complement synthesis (C1q) and complement deposition (C3b and C5b-9), T-cell exhaustion, and extensive tissue damage (Nienhold et al, 2020).

Among the treatments assayed for COVID-19, the use of immunomodulators has shown promising results. Dexamethasone and IL-6 blocking agents have exhibited their usefulness and are currently used in the treatment of severe COVID-19–affected individuals (The RECOVERY Collaborative Group et al, 2021; The REMAP-CAP, 2021). Moreover, other treatments based on anti-complement drugs are still under clinical evaluation with encouraging results (AMY-101, vilobelimab, and eculizumab) (Polycarpou et al, 2020; Kurtovic & Beeson, 2021).

[1]IrsiCaixa AIDS Research Institute, Germans Trias i Pujol Research Institute (IGTP), Can Ruti Campus, Badalona, Spain  [2]Infectious Diseases Department, Fight Against AIDS Foundation (FLS), Germans Trias i Pujol Hospital, Badalona, Spain  [3]Chair in Infectious Diseases and Immunity, Centre for Health and Social Care Research (CESS), Faculty of Medicine. University of Vic–Central University of Catalonia (UVic–UCC), Vic, Spain  [4]Universitat Autonoma de Barcelona, Cerdanyola Del Vallès, Spain  [5]CIBERes: Centro de investigaciones en Red de Enfermedades Respiratorias Del Instituto Carlos III, Madrid, Spain

Correspondence: jcarrillo@irsicaixa.es; mlpbotet.germanstrias@gencat.cat
*María Luisa Rodríguez de la Concepción, Erola Ainsua-Enrich, and Esteban Reynaga contributed equally to this work

Intravenous immunoglobulins (IVIGs) are intensively used for the treatment of immunodeficient individuals as "replacement therapy." However, at high dose (1–2 g/kg), they are also used as "immunomodulatory agents" for some autoimmune diseases. IVIG can reduce inflammation by several mechanisms which include the development of T-regulatory cells, the reduction of autoantibodies and cytokines levels, the prevention of the activation of innate immune cells, or by blocking the activation of the complement cascade (Schwab & Nimmerjahn, 2013). We and others have shown the success in the treatment of COVID-19 using high dose of IVIG (Cao et al, 2020; Gharebaghi et al, 2020; Lanza et al, 2020; Reynaga et al, 2020; Xie et al, 2020). Nevertheless, how IVIG can help in the recovery of COVID-19 is not well understood. Here, we have shown that IVIG may reduce the levels of several soluble factors involved in immune cell recruitment and activation. In addition, IVIG could reduce the levels of C5a in plasma, a powerful inflammatory mediator, generated by C5 cleavage during complement activation, which plays a major role recruiting and activating immune cells.

# Results and Discussion

Previously, we have shown that the administration of high-dose of IVIG to five COVID-19 patients was well tolerated and improved their recovery, normalizing the hyperinflammation status and restoring the levels of lymphocytes (Reynaga et al, 2020). At the start of treatment, they all showed a $PaO_2/FiO_2$ values ≤300 mmHg and multilobar pneumonia and were receiving supplemental oxygen therapy. All these clinical parameters improved notoriously after treatment initiation (Reynaga et al, 2020). Here, we aimed to characterize how IVIG reduced inflammation in these individuals. Thus, we performed an analysis of 41 soluble biomarkers in plasma by Luminex and ELISA (Table S1). Whereas most of these biomarkers were not consistently altered during treatment (Figs S1 and S2), we have observed a descendant trend ($P$ = 0.062, Wilcoxon matched-pairs signed rank test) in the levels of some soluble proteins that might support the clinical benefit observed in IVIG-treated patients (Figs 1 and 2). Unfortunately, the low number of IVIG-treated patients did not allow us to find statistically significant differences.

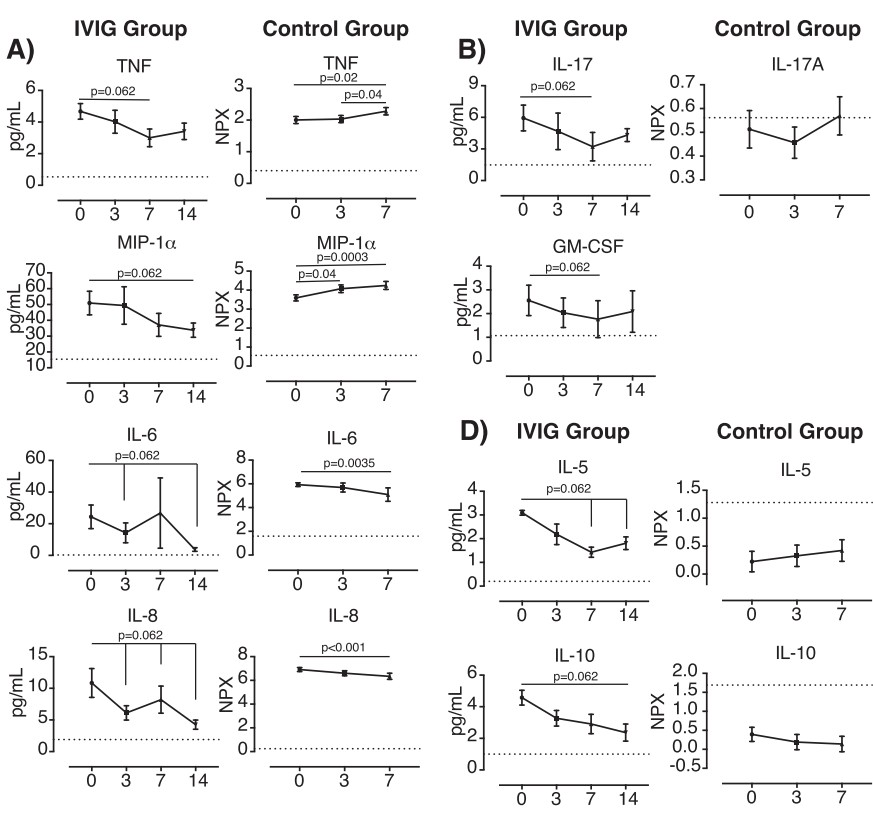

**Figure 1.  Th1, Th12, and Th17 biomarkers and sCD163 in plasma samples from IVIG-treated COVID-19+ individuals.**
**(A, B, C, D)** Levels of (A) TNF, macrophage inflammatory protein-1α, IL-6, and IL-8; (B) IL-17A and GM-CSF (C) sCD163 and (D) IL-5 and IL-10, were determined in plasma samples from high-dose IVIG-treated COVID-19+ individuals (n = 5) (IVIG group) by Luminex or ELISA at baseline (day 0) and days 3, 7, and 14 since treatment initiation. Publicly available data showing the levels of these molecules in plasma of IVIG-untreated COVID-19 patients (control group) (n = 32) at enrollment (day 0) and days +3 and +7 were used for comparative purposes (https://www.olink.com/mgh-covid-study). Mean and standard error of the mean are shown. Data of IVIG group were analyzed using the Wilcoxon matched-pairs signed rank test. Data in the control group were analyzed using Friedman and Dunn's multiple comparisons test. NPX, normalized protein expression. Dotted lines indicate the limit of detection.

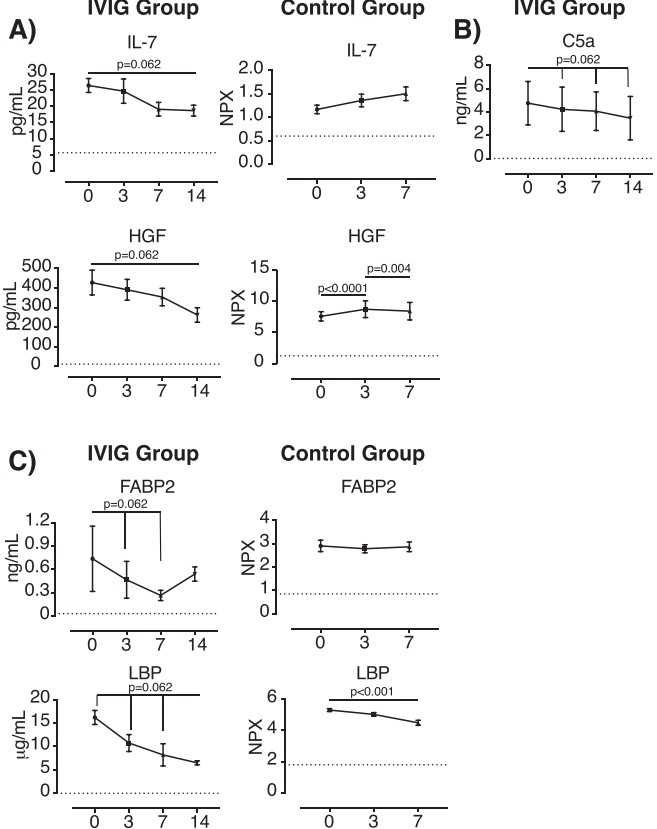

**Figure 2. Effect of IVIG treatment in the plasmatic levels of IL-7, hepatocyte growth factor, fatty acid–binding protein 2, and C5a in COVID-19 patients.**
**(A, B, C)** Levels of (A) IL-7, hepatocyte growth factor; (B) C5a complement factor; and (C) fatty acid–binding protein 2, and LPS-binding protein were determined in plasma samples from high-dose IVIG-treated COVID-19+ individuals (n = 5) (IVIG group) by Luminex or ELISA at baseline (day 0) and days 3, 7 and 14 after treatment initiation. Publicly available data showing the levels of these molecules in plasma of IVIG-untreated COVID-19 patients (control group) (n = 32) at enrollment (day 0) and days +3 and +7 were used for comparative purposes (https://www.olink.com/mgh-covid-study). Mean and Standard Error of the Mean are shown. Data of IVIG group were analyzed using the Wilcoxon matched-pairs signed rank test. Data in the control group were analyzed using Friedman and Dunn's multiple comparison test. NPX, normalized protein expression. Dotted lines indicate the limit of detection.

Publicly available data describing the levels of these molecules in plasma of COVID-19 patients were used for comparative purposes (https://www.olink.com/mgh-covid-study) (Filbin et al, 2021) (Figs 1 and 2). Several proinflammatory factors, such as IL-6 (previously reported [Reynaga et al, 2020]), IL-8, tumor necrosis factor (TNF), and macrophage inflammatory protein-1α (MIP-1α), showed a descendant trend in the IVIG-treated group (Fig 1A). However, the opposite behavior was observed for TNF and MIP-1α during the natural course of the disease in COVID-19 patients (Fig 1A). The data also suggested that the IVIG might induce a faster decade in the plasmatic levels of IL-6 and IL-8 (day 3 after treatment initiation) when compared with the reduction observed in the control group (day 7) (Fig 1A).

In addition, high levels of IL-17A and GM-CSF have been described in both sera and bronchoalveolar fluid of severe COVID-19 patients, and their levels were associated with the expansion in the lung of pathogenic tissue-resident memory-like Th17 cells (Zhao et al, 2021). Interestingly, the levels of IL-17 and GM-CSF tended to decrease in IVIG-treated COVID-19 patients until day 7 since IVIG treatment initiation (Fig 1B). Although, IL-17A was hardly detected in the control group and no data of GM-CSF were available, the decreasing trend observed in IVIG-treated COVID-19 patients suggests that IVIG might modulate also the Th17 response (Maddur et al, 2011).

However, no effect of IVIG treatment was observed on the levels of IFN-γ–related biomarkers (Fig S2). Despite IFN-γ was hardly detected in IVIG-treated patients, a reduction in the levels of CXCL-10 was found in both IVIG-treated and control groups. In contrast, the levels of CXCL-9 did not change in any group (Fig S2).

Soluble CD163 is considered a marker of monocyte/macrophage activation, and high levels of soluble CD163 have been linked to several pathologic conditions, such as infectious disease or autoimmunity (Ingels et al, 2013). Accordingly, sCD163 is increased in plasma of SARS-CoV-2–infected individuals and has been associated with the progression to severe COVID-19 (Gómez-Rial et al, 2020; Zingaropoli et al, 2021). Although IVIG-treated COVID-19 patients showed an improvement on their clinical status that was associated with a decreasing trend in several biomarkers related to innate cells activation and recruitment, the levels of sCD163 showed a rapid increasing tendency (as early as 3 d after the first administration of IVIG) (Fig 1C). However, this increase was only observed at day 7 but not at day 3 in IVIG-untreated patients (Fig 1C). An increase in the plasmatic levels of sCD163 was also observed in a subgroup of patients affected by Kawasaki disease that positively responded to IVIG treatment (Azuma et al, 2020). Therefore, the levels of sCD163 observed in IVIG-treated COVID-19 patients might indicate an IVIG-dependent modulation of macrophage/monocyte function and should not be considered as a biomarker of disease severity.

In addition to proinflammatory cytokines and chemokines, the Th2 cytokines IL-5 and IL-10 also showed a decreasing trend after the administration of high dose of IVIG ($P = 0.062$); however, the effect of IVIG treatment could not be confirmed because these cytokines were below the limit of detection in the untreated control group (Fig 1D). Furthermore, the levels of IL-7 and hepatocyte growth factor (HGF), two key factors involved in lymphocyte generation and expansion, and tissue remodeling and wound healing, respectively, also tended to decrease after IVIG treatment initiation ($P = 0.062$) (Fig 2A). These results contrasted with the dynamic of these proteins observed in untreated COVID-19 patients, who showed an increase in the levels of those proteins over time. Interestingly, it has been described that HGF plays a major role in the expansion of myeloid-derived suppressor cells (Yen et al, 2013). These cells show potent immunosuppressor properties, are expanded in severe COVID-19 patients, and their frequency correlate with the levels of HGF (Kvedaraite et al, 2021). Moreover, high levels of HGF have been associated with severity and mortality in SARS-CoV-2–infected individuals (Tamayo-Velasco et al., 2021).

Complement activation has been also associated with tissue damage, hyperinflammation, and disease severity in COVID-19 individuals (Carvelli et al, 2020; Java et al, 2020; Shen et al, 2020). High levels of C5a have been associated with the severity of the disease in COVID-19 patients (Carvelli et al, 2020; Cugno et al, 2020), and several therapies aimed to reduce complement activation are in

clinical trials (Polycarpou et al, 2020). Furthermore, other highly pathogenic viruses, such as H5N1, H7N9, SARS-CoV, or MERS-CoV, might also induce acute lung injury by an excessive and uncontrolled complement activation (Wang et al, 2015). Here, we have shown that the levels of C5a showed a descendant trend in plasma after treatment initiation ($P$ = 0.062) (Fig 2B), suggesting that IVIG may reduce the levels of C5a in COVID-19 subjects. Although we do not have access to data from untreated COVID-19 patients to confirm this descendant trend, it has been reported that the levels of C5a remains stable at least during 10 d after disease initiation (Carvelli et al, 2020). In fact, among the different immunomodulatory mechanisms linked to high-dose IVIG treatment, the binding and blocking of C5a and C3a have been well documented (Basta et al, 2003). Therefore, the trend observed in the levels of C5a in the IVIG-treated group is, at some extend, expected, and might be considered as a positive control of the immunomodulatory effect of the IVIG in COVID-19 patients. Interestingly, we did not observe a reduction in the levels of sC5b-9 (Fig S1), suggesting that the formation of the membrane attack complex might not be affected. Similar results were observed when specific reagents for blocking C5a or its receptor (C5aR1) were used (Carvelli et al, 2020). C5a is a potent proinflammatory and chemoattractant factor for neutrophils, monocytes, and macrophages (Guo & Ward, 2005). C5a increases the production of TNF, MIP-1α, and IL-6 by monocytes isolated from COVID-19 patients and stimulated with LPS or R848 (Carvelli et al, 2020). Interestingly, all these proteins tend to be reduced in IVIG-treated COVID-19 patients.

Overall, our results suggest that IVIG could decrease the hyperinflammation stage observed in SARS-CoV-2–infected individuals by reducing the levels of cytokines, chemokines and some complement factors (C5a) involved in immune activation and cell migration. Interestingly, most of the biomarkers described in the present work had been reported to be elevated in severe COVID-19 individuals (Blanco-Melo et al, 2020; Huang et al, 2020).

However, IVIG are multifunctional and they might also act through additional mechanisms that need further investigation. In this sense, we have also observed a progressive reduction in the levels of fatty acid–binding protein 2 (FABP-2) and LPS-binding protein (LBP) after IVIG treatment initiation (Fig 2C). Plasmatic FABP-2 has been considered as a biomarker associated with gut damage (Pelsers et al, 2005), and a plasmatic increase in the levels of endotoxin (1 → 3)-β-d-glucan and bacterial DNA (markers of gut leakage) have been also described in severe COVID-19 patients (Sirivongrangson et al, 2020). About 12% of COVID-19 patients manifest gastrointestinal symptoms whereas SARS-CoV-2 shedding in feces is observed in 40% of infected individuals (Parasa et al, 2020). Therefore, the decreasing trend observed in FABP-2 and LBP might indicate that high-dose IVIG might reduce gut damage in COVID-19 patients. However, this finding needs further investigation to clarify the role of these biomarkers in the pathogenesis of COVID-19 and how high-dose IVIG treatment might modulate their functions. Guedj K et al have recently reported that the levels of FABP2 are lower in COVID-19 patients than in uninfected individuals, and are not associated with prognosis (Guedj et al, 2021). Moreover, despite it had been shown that COVID-19 patients may have high levels of LBP in serum (Hoel et al, 2020), the plasmatic levels of this molecule are reduced in both IVIG-treated and untreated COVID-19 patients over time (Fig 2C). In spite of that, IVIG-treated individuals might show a faster decreasing trend because it was observed as early as day 3 compared with day 7 in the control group (Fig 2C). LBP plays a major role in LPS signaling. It binds to LPS and forms a

complex with CD14 that finally induces the endocytosis and signaling through TLR4 (Tsukamoto et al, 2018), promoting a proinflammatory cascade. It has been reported that the spike glycoprotein of SARS-CoV-2 can bind to LPS and promote inflammation in vitro and in vivo (Petruk et al, 2020). Moreover, the exposure of pigs to both porcine respiratory coronavirus and LPS enhanced respiratory disease by increasing the proinflammatory response (Van Reeth et al, 2000). Therefore, it is plausible to think that the LPS pathway might be involved in the development of the severe COVID-19 (Kruglikov & Scherer, 2021) and that IVIG treatment might modulate this pathway. Nevertheless, no major effect on soluble CD14 was observed (Fig S1), which remained at normal levels in IVIG-treated individuals.

The present work shows two main limitations: the low number of individuals analyzed and the lack of specific controls. To overcome the later limitation, we have used publicly available data obtained from COVID-19 patients. We have selected individuals that showed a similar disease profile than the IVIG group. Therefore, our results suggest that high-dose IVIG administration may be a promising treatment for, at least, a subset of COVID-19–affected individuals. However, because of the limitations described before, the results presented here need to be confirmed in future studies. A randomized clinical trial is currently ongoing to corroborate our preliminary finding (clinical trial.gov: NCT04432324).

## Materials and Methods

### Samples

Blood samples were collected in EDTA tubes from five COVID-19 patients successfully treated with a 5-d course of 400 mg/kg/d of IVIG (Flebogamma 10% DIF; Instituto Grifols). Blood samples were obtained at four different time points: Day 0 (pretreatment initiation), 3, 7, and 14 after the first administration of IVIG. Blood samples were processed within 3 h of collection. To collect plasma, blood samples were centrifuged at 3,000$g$ for 10 min at room temperature. Plasma samples were stored at –80°C until use. Clinical and demographic features of patients have been previously reported (Reynaga et al, 2020). Briefly, patients showed a multilobar pneumonia and $PaO_2$/$FiO_2$ values ≤300 mmHg. All required supplemental oxygen therapy and showed a medium risk of acute acute respiratory distress syndrome according to the Berlin score. SARS-CoV-2 infection was confirmed by RT-PCR in oropharyngeal swab samples (four out of five patients) or by serology (one out of five patients). Their ages were in the range of 24–80 yr and two of them were female. The study was revised and approved by the Pharmacy and Therapeutics Committee of the "Hospital Germans Trias I Pujol" (code: PI-20-266) and conformed to the principles set out in the WMA Declaration of Helsinki. A signed informed consent was obtained for all participants.

### Control group

Publicly available longitudinal data from COVID-19 patients were used as controls (https://www.olink.com/mgh-covid-study) (Filbin et al, 2021). The control group (n = 32) was selected according to the following criteria: COVID status: positive; age cat: 1 (20–34 yr old), 2 (36–49), 4 (65–79), and 5 (>80); respiratory symptoms: positive (sore throat, congestion, productive or dry cough, shortness of breath or

hypoxia, or chest pain); World Health Organization score basal 0: 4 (hospitalized, supplementary O2 required); C-reactive protein at day 0: 3 (0–19.9), 4 (100–179), or 5 (>180). Only individuals with data at day 0, 3, and 7 were considered.

### Determination of biomarkers in plasma sample

The concentration of several soluble biomarkers was determined in plasma samples by Luminex or ELISA. A description of the molecules determined, suppliers and kits used for the determination are showed in Table S1. All determinations were performed following the manufacturers' instructions, except for the incubation of samples in ELISA assays, which was performed at 4°C, overnight. For ELISA development, the supersensitive 3, 3′, 5, 5′-tetramethylbenzidine (TMB) Liquid Substrate (T4444; Sigma-Aldrich) was used. Reaction was stopped using 2 M of $H_2SO_4$ and signal was analyzed as 450/620 ratio.

### Statistical analysis

The levels of each biomarker at day 0, 3, 7, or 14 in the IVIG group were analyzed using the Wilcoxon matched-pairs signed rank test. The control group was analyzed using Friedman and Dunn's multiple comparison tests. Statistical analyses were conducted using GraphPad Prism v 7.0.

## Data Availability

Data generated in the present study are described in the manuscript and supplementary materials. The full data set is available at 10.6084/m9.figshare.14925219.

## Supplementary Information

## Acknowledgements

We would like to thank all the patients who participated in this study, and all the people that are supporting our research through the "#YomeCorono" crowdfunding initiative. C Ávila-Nieto is supported by the predoctoral grant 2020 FI_B_0742 from "Generalitat de Catalunya" and "European Social Fund." We would like to thank Grifols that provided free of charge all the IVIG required for this study. Grifols had no role on the study design or decision to publish. We would also like to acknowledge the data provided by the Massachusetts General Hospital Emergency Department COVID-19 Cohort (Filbin, Goldberg, and Hacohen) with Olink Proteomics.

### Author Contributions

ML Rodriguez de la Concepcion: data curation, formal analysis, investigation, methodology, and writing—review and editing.
E Ainsua-Enrich: data curation, formal analysis, investigation, methodology, and writing—review and editing.
E Reynaga: conceptualization, validation, investigation, and writing—review and editing.
C Avila-Nieto: data curation, investigation, methodology, and writing—review and editing.
JR Santos: investigation and writing—review and editing.
S Roure: investigation and writing—review and editing.
L Mateu: investigation and writing—review and editing.
R Paredes: investigation and writing—review and editing.
J Puig: data curation, investigation, and writing—review and editing.
JM Jimenez: data curation and writing—review and editing.
N Izquierdo-Useros: investigation and writing—review and editing.
B Clotet: conceptualization, supervision, funding acquisition, investigation, visualization, and writing—review and editing.
ML Pedro-Botet: conceptualization, data curation, investigation, and writing—review and editing.
J Carrillo: data curation, formal analysis, supervision, validation, investigation, methodology, and writing—original draft.

### Conflict of Interest Statement

B Clotet is the founder of AlbaJuna Therapeutics SL. J Carrillo is the cofounder and Chief Scientific Officer of AlbaJuna Therapeutics SL. B Clotet, J Carrillo, and N Izquierdo-Useros are part of the CBIG consortium that is founded by Grifols. The authors declare no other conflict of interests.

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
