## [Reviewer comments · Life Science Alliance]

Life Science Alliance

High-dose intravenous immunoglobulins might modulate inflammation in COVID-19 patients.

Maria Luisa Rodriguez de la Concepcion, Erola Ainsua Enrich, Esteban Alberto Reynaga Sosa, Carlos Avila-Nieto, Jose Ramon Santos, Silvia Roure, Lourdes Mateu, Roger Paredes, Jordi Puig, Juan Manuel Jimenez, Nuria Izquierdo-Useros, Bonaventura Clotet, Maria Luisa Pedro-Botet, and Jorge Carrillo

DOI: <https://doi.org/10.26508/lsa.202001009>

Corresponding author(s): Jorge Carrillo, IrsiCaixa Institute for AIDS Research and Maria Luisa Pedro-Botet, Infectious Diseases Department, Fight against AIDS Foundation (FLS), Germans Trias i Pujol Hospital, Badalona, Catalonia, Spain

Review Timeline:

Submission Date:	2020-12-30
Editorial Decision:	2021-03-22
Revision Received:	2021-06-22
Editorial Decision:	2021-07-01
Revision Received:	2021-07-08
Accepted:	2021-07-12

Transaction Report:

March 22, 2021

Re: Life Science Alliance manuscript #LSA-2020-01009

Dr. Jorge Carrillo
IrsiCaixa Institute for AIDS Research
Ctra del Canyet s/n
Hospital Universitario Germans Trias i Pujol
Badalona, Barcelona 08912
Spain

Dear Dr. Carrillo,

Thank you for submitting your manuscript entitled "High-dose MIG decrease the levels of cytokines, complement activation and LBP in COVID-19 patients" to Life Science Alliance. The manuscript was assessed by expert reviewers, whose comments are appended to this letter.

We apologize for this unusual and extended delay in getting back to you. As you will note from the reviewers' comments below, the reviewers' enthusiasm for this study is mixed. Reviewer 1 thinks that there is merit to revising the study, reviewer 2 is concerned about the small sample size, which affects the statistics. While we appreciate the points raised by Reviewer 2, we understand that increasing the sample size would be difficult and can overrule that request. We encourage you to resubmit a revised manuscript back to LSA that addresses all of Reviewer 1's points including the request for analyzing publicly available longitudinal data and provide controls, clarifications and text changes in response to the Reviewer 2's points.

Thank you for this interesting contribution to Life Science Alliance. We are looking forward to

receiving your revised manuscript.

Sincerely,

Shachi Bhatt, Ph.D.

Executive Editor

Life Science Alliance

<https://www.lsjournal.org/>

Interested in an editorial career? EMBO Solutions is hiring a Scientific Editor to join the international Life Science Alliance team. Find out more here -

https://www.embo.org/documents/jobs/Vacancy_Notice_Scientific_editor_LSA.pdf

B. MANUSCRIPT ORGANIZATION AND FORMATTING:

Reviewer #1 (Comments to the Authors (Required)):

The authors longitudinally measured soluble factors in five COVID-19 patients treated with IVIG. The data is interesting and important for other scientists designing similar studies. The reviewer has some comments/suggestions:

1. The reviewer would like to draw the authors attention to another major limitation of the study, namely the absence of a control group. Without a control group it is thus not clear if the findings detected here are related to IVIG treatment or just represent a natural course of disease in the studied individuals. To overcome this, publically available longitudinal data could be used, here is one example: <https://www.olink.com/mgh-covid-study/>. Although matching might be challenging, the authors in this way may still get a useful reference for what represents a natural course of COVID-19 and which changes might actually be induced by the IVIG treatment in the present study.

2. The authors discuss the decrease of measured markers, and although trends were detected, they did not reach statistical significance threshold normally used (i.e. p value <0.05). The reviewer is not concerned by that per se, and the detected findings do not become less interesting just because of that. However, the reviewer would encourage the authors to use more careful presentation of the findings, trying to avoid overstatements (including the title), and talk more about the hypothesis generating results rather than definitive evidence.

Minor comments:

1. The authors mention sHLH, which of course might occur in COVID-19 and was at the centre of debate of COVID-19 associated hyperinflammation in the beginning. However, the current understanding is that sHLH is very rare in COVID-19. And the authors may also agree that the hyperinflammatory state, also combined with systemic thrombogenicity, in COVID-19 is rather different from the one seen in sHLH or MAS. Therefore, if the authors wishes to keep sHLH in the picture, the reviewer would advise a more balanced presentation.

2. In terms of immune related therapies, the reviewer suggests that there are the two major treatments that so far showed efficacy in large studies, one, also mentioned by the authors is dexamethasone (PMID: 32678530), and another one is IL-6 blockage: doi: <https://doi.org/10.1101/2021.01.07.21249390>. The authors may consider mentioning both.

3. The authors state that from the 41 markers analyzed: "...most of these biomarkers were not consistently altered during treatment (data not shown)". The reviewer suggests to also include this data that is "not shown" in the supplement, as it might be helpful for other physician scientists while planning/conceiving similar studies.

4. HGF, also known to support myeloid derived suppressor cell expansion (PMID: 24052949), was in COVID-19 previously shown to correlate with clusters of myeloid-derived suppressor like cells (PMID: 33479167). As the expansion of those cells is one of the hallmarks of severe COVID-19, it is indeed interesting that the authors detect a trend towards decrease of HGF after initiation of IVIG.

5. For the curiosity of the reviewer, which soluble factors showed best correlation with HGF?

6. The changes of one marker related to complement activation, namely C5a, alone perhaps does not represent activation of the whole complement cascade, and therefore, here it sounds more as an overstatement. The reviewer therefore advises to tone this down through the text.

7. The reviewer assumes that diagnosing COVID-19 in patients included PCR positivity for SARS-

CoV-2? The authors may consider mentioning this in the text.

8. Title: the reviewer advises to spell out LBP (if it is felt the title then gets too long, "LPS binding protein" could be an option).

Reviewer #2 (Comments to the Authors (Required)):

The authors measured 41 cytokines in plasma of patients with severe COVID-19 prior and after high-dose of intravenous immunoglobulins (IVIG). Among these 41 cytokines authors claims that 12 of them displayed a decrease suggesting that IVIG could be use on Hospitalized severe COVID-19 patients.

Sadly, the results presented in this manuscript are weak, only 5 patients. Due to this low number none of the statistical test are significant so the authors can only state a trend for all the cytokines presented in the 3 figures. By consequence, I do not believe authors should made any conclusion on the treatment effectivity. No data are shown on patients not receiving IVIG so the decreased observed could just be due to a decrease of the inflammation process. Moreover, the authors elaborate hypothesis about LPS but it has never been shown that LPS is increased in blood circulation of severe COVID-19 or at least they didn't mention the reference. The authors are missing many crucial references in their article Guo et al., 2020; Schulte-Schrepping et al., 2020; Silvin et al., 2020; Wen et al., 2020; Zhang et al., 2020; Mathew et al., 2020; Su et al., 2020). I do not need to remind the authors that the same kind of claim as been done with many other molecules like the famous case of Hydroxychloroquin. It is important to have controls and also read the fundamental articles. For exemple, LBP upregulation could be induced by the interaction of calprotectin and TLR4 (TLR4 being also LPS receptor), IVIG treatment could interfere with this interaction and by consequence induces a decrease of LBP without any LPS around. However, the data presented are too weak to explore any hypothesis.

Reviewer #1 (Comments to the Authors (Required)):

The authors longitudinally measured soluble factors in five COVID-19 patients treated with IVIG. The data is interesting and important for other scientists designing similar studies. The reviewer has some comments/suggestions:

1. The reviewer would like to draw the authors attention to another major limitation of the study, namely the absence of a control group. Without a control group it is thus not clear if the findings detected here are related to IVIG treatment or just represent a natural course of disease in the studied individuals. To overcome this, publically available longitudinal data could be used, here is one example: <https://www.olink.com/mgh-covid-study/>. Although matching might be challenging, the authors in this way may still get a useful reference for what represents a natural course of COVID-19 and which changes might actually be induced by the IVIG treatment in the present study.

We agree with the reviewer that the lack of a control group is a weakness of the present work. We would like also to thank to the reviewer for his/her constructive comment regarding the use of publicly available data to confirm whether the trends observed in our study might be due to the IVIG treatment or due to the natural course of the disease. Following the reviewer's advice, we have analyzed the natural dynamics of the selected parameters using the data available at: <https://www.olink.com/mgh-covid-study>

In the new Fig 1 and Fig 2, we have included both analyses for comparative purposes. Only TNF, IL-10, IL-7, MIP-1 α , HGF, IL17A and FABP-2 showed a distinctive behavior. Whereas we observed a descendant trend in the levels of TNF, MIP-1 α , IL-7, and HGF in IVIG-treated patients, the levels of these biomarkers increased in untreated COVID-19 patients overtime. These results support the notion that IVIG treatment might have a positive effect on the control of immune activation in COVID-19 patients. In addition, we have also observed a different dynamics in the levels of sCD163 and IL6 (Fig 1). Although no data from GM-CSF were available in <https://www.olink.com/mgh-covid-study> for comparative purposes, the increase of GM-CSF and IL-17A in COVID-19 patients and their association with severity are well documented in the literature (Zhao et al. *Sci. Immunol.* 6. doi:10.1126/SCIIMMUNOL.ABF6692). We have observed a decreasing trend in the levels of IL17A and GM-CSF that suggest that IVIG could modulate the Th17 response as has been described previously (Maddur et al *J. Allergy Clin. Immunol.* 127:823-830.e7. doi:10.1016/j.jaci.2010.12.1102). Moreover, IL17A, IL10 and IL-5 also showed a decreasing trend in IVIG treated individuals. Unfortunately, the levels of these cytokines in the control group were below the limit of detection, so no formal conclusion can be taken from these molecules.

IL-8 seems to decrease in both IVIG treated and untreated COVID-19 individuals so we cannot conclude that IVIG treatment is behind the reduction in IL-8. However, the decreasing trend observed in IVIG treatment seemed to be more rapid than the one observed in the control group, suggesting that IVIG might accelerate this process.

Levels of C5a were also not available in <https://www.olink.com/mgh-covid-study>. However, it has been reported that the levels of C5a were associated with the severity of the disease and that high levels of C5a remained stable at least during 10 days (Carvelli J. et al. Nature. 588:146–150. doi:10.1038/s41586-020-2600-6). Our results show that the levels of C5a progressively decrease overtime. These data agree with the capability of IVIG of binding and neutralize C5a (Basta M. et al. 2003 Nat Med DOI: 10.1038/nm836).

The decreasing trend observed in the levels of LBP in IVIG-treated COVID-19 patients was also observed in publicly available data obtained from untreated individuals (Fig 2C). Therefore, we cannot claim that IVIG reduces the levels of LBP and we have removed this notion from the title. However, similarly to IL-8, the decrease in the levels of LBP seems to be accelerated by the IVIG treatment. Interestingly, we also observed a decreasing trend in the levels of Fatty Acid Binding Protein 2, a biomarker of intestinal damage whose reduction was not observed in the control group. The significance of these results is not clear since it has been reported that COVID-19 patients showed lower levels of FABP2 in plasma than healthy controls and that the levels of LBP were also reduced overtime in the control group. However, it has been reported that severe COVID-19 could be associated with a gut leakage and that LPS could contribute to the hyperinflammatory status described in these patients. Accordingly, we have now included this discussion in the new version of the manuscript.

2. The authors discuss the decrease of measured markers, and although trends were detected, they did not reach statistical significance threshold normally used (i.e. p value <0.05). The reviewer is not concerned by that per se, and the detected findings do not become less interesting just because of that. However, the reviewer would encourage the authors to use more careful presentation of the findings, trying to avoid overstatements (including the title), and talk more about the hypothesis generating results rather than definitive evidence.

Again, we totally agree with the reviewer. We have tried to smooth the tone and avoid overstatements. Moreover, we have simplify the figures so the most relevant data are captured in Fig 1 and 2.

Minor comments:

1. The authors mention sHLH, which of course might occur in COVID-19 and was at the centre of debate of COVID-19 associated hyperinflammation in the beginning. However, the current understanding is that sHLH is very rare in COVID-19. And the authors may also agree that the hyperinflammatory state, also combined with systemic thrombogenicity, in COVID-19 is rather different from the one seen in sHLH or MAS. Therefore, if the authors wishes to keep sHLH in the picture, the reviewer would advise a more balanced presentation.

As the reviewer notes, the manifestation of sHLH in severe COVID-19 is a point of debate. In order to avoid this and because sHLH is not the main focus of the manuscript, we have removed any reference to the sHLH. Thus, the new sentence reads as follows:

“ An early and sustained increase in cytokine and chemokines levels has been associated with severe cases of the disease (Huang et al., 2020; Lucas et al., 2020)”

2. In terms of immune related therapies, the reviewer suggests that there are the two major treatments that so far showed efficacy in large studies, one, also mentioned by the authors is dexamethasone (PMID: 32678530), and another one is IL-6 blockage: doi: <https://doi.org/10.1101/2021.01.07.21249390>. The authors may consider mentioning both.

We have added the information related to IL-6 blockade:

“Among the treatment assayed for COVID-19, the use of immunomodulators has shown promising results. Among them, Dexamethasone and IL-6 blocking agents have shown their usefulness and are currently used in the treatment of severe COVID-19 affected individuals (The REMAP-CAP, 2021; The RECOVERY Collaborative Group, 2020)”

3. The authors state that from the 41 markers analyzed: "...most of these biomarkers were not consistently altered during treatment (data not shown)". The reviewer suggests to also include this data that is "not shown" in the supplement, as it might be helpful for other physician scientists while planning/conceiving similar studies.

This information has been added to the manuscript as Fig S1.

4. HGF, also known to support myeloid derived suppressor cell expansion (PMID: 24052949), was in COVID-19 previously shown to correlate with clusters of myeloid-derived suppressor like cells (PMID: 33479167). As the expansion of those cells is one of the hallmarks of severe COVID-19, it is indeed interesting that the authors detect a trend towards decrease of HGF after initiation of IVIG.

We would like to thank to the reviewer for the constructive comments. We have included this information in the manuscript for discussing the putative role of the reduction in the levels of HGF that we have observed.

“Interestingly, it has been described that HGF plays a major role in the expansion of myeloid-derived suppressor cells(Yen et al., 2013). These cells show potent immunosuppressor properties, are expanded in severe COVID-19 patients and their frequency correlate with the levels of HGF(Kvedaraitė et al.,

2021). Moreover, HGF has been associated with severity and mortality in SARS-CoV-2 infected individuals (Tamayo-velasco et al., 2021).”

5. For the curiosity of the reviewer, which soluble factors showed best correlation with HGF?

We have performed a correlation analysis among the HGF and other biomarkers that showed a decreasing trend after IVIG treatment. The strongest correlation was observed with IL-6 and IL-10. It has been described that HGF suppress the expression of IL-6 by LPS stimulated BM-derived macrophage whereas enhance the production of IL-10. That could explain the association observed between HGF and IL-6 and IL-10.

6. The changes of one marker related to complement activation, namely C5a, alone perhaps does not represent activation of the whole complement cascade, and therefore, here it sounds more as an overstatement. The reviewer therefore advises to tone this down through the text.

We have modified the text, even in the title, to avoid this overstatement.

7. The reviewer assumes that diagnosing COVID-19 in patients included PCR positivity for SARS-CoV-2? The authors may consider mentioning this in the text.

The following information has been added to the material and methods section: SARS-CoV-2 infection was confirmed by RT-PCR in oropharyngeal swab samples (4 out of 5 patients) or by serology (1 out of 5 patients) Reynaga E, et al. (2020).

8. Title: the reviewer advises to spell out LBP (if it is felt the title then gets too long, "LPS binding protein" could be an option).

To simplify the title, we have removed LBP; however, following the suggestion of the reviewer, LBP and other acronyms have been spelled out through the text.

Reviewer #2 (Comments to the Authors (Required)):

The authors measured 41 cytokines in plasma of patients with severe COVID-19 prior and after high-dose of intravenous immunoglobulins (IVIG). Among these 41 cytokines authors claims that 12 of them displayed a decrease suggesting that IVIG could be use on Hospitalized severe COVID-19 patients.

Sadly, the results presented in this manuscript are weak, only 5 patients. Due to this low number none of the statistical test are significant so the authors can only state a trend for all the cytokines presented in the 3 figures. By consequence, I do not believe authors should made any conclusion on the treatment effectivity. No data are shown on patients not receiving IVIG so the decreased observed could just be due to a decrease of the inflammation process. Moreover, the authors elaborate hypothesis about LPS but it has never been shown that LPS is increased in blood circulation of severe COVID-19 or at least they didn't mention the reference. The authors are missing many crucial references in their article Guo et al., 2020; Schulte-Schrepping et al., 2020; Silvin et al., 2020; Wen et al., 2020; Zhang et al., 2020; Mathew et al., 2020; Su et al., 2020). I do not need to remind the authors that the same kind of claim as been done with many other molecules like the famous case of Hydroxychloroquin. It is important to have controls and also read the fundamental articles. For example, LBP upregulation could be induced by the interaction of calprotectin and TLR4 (TLR4 being also LPS receptor), IVIG treatment could interfere with this interaction and by consequence induces a decrease of LBP without any LPS around. However, the data presented are too weak to explore any hypothesis.

We would like to thank to the reviewer for the time dedicated to revise our manuscript. Moreover, we agree with the reviewer in that the limited number of patients included in the present work only allow us to identify some trends and no robust conclusions can be taken. This is a limitation of our work that we have highlighted in the text. Despite that, the comparison with publicly available data, that evaluate the evolution of several biomarkers in the course of the disease without IVIG treatment (as recommended by the reviewer 1), has now shown that the decreasing trend observed for some biomarkers in the IVIG treated group (TNF, MIP-1 α , IL-6, IL-8, IL-5, IL-10, IL-17A, GM-CSF, FABP2, LBP, sCD163 and C5a) might differ from the one observed in untreated individuals. Therefore, it is plausible to think that IVIG treatment can promote the modulation of these molecules. Interestingly, high levels of most of those biomarkers have been identified in severe COVID-19 cases, and some of them have been linked to bad prognosis (i.e HGF) (J.Clin. Med. 2021, 10, 2017)

IVIG are multifunctional, and their immunomodulatory properties can be linked to different pathways, which involve both Fc and Fab-mediated functions. Our results suggest that high dose of IVIG might modulate the hyperinflammation and immunosuppression status observed in COVID-19 patients. It is unlikely that this effect

is random since many biomarkers are not altered (Fig S1) and others follow the natural kinetics observed in untreated COVID-19 patients (IFN- α , IP-10, MIG). On the contrary, the decreasing trends observed for several soluble factors in our work seem to be very specific of several pathways involving, for instance, the recruitment and activation of innate and Th17 immune cells. Among the different immunomodulatory mechanisms linked to high dose IVIG, the binding and blocking of C5a and C3a has been well documented (Basta M et al. 2003. *Nature Medicine*. doi:10.1038/nm836). Therefore, the trend observed in the levels of C5a in the IVIG treated group is, at some extent, expected, and might be considered as a positive control of the immunomodulatory effect of the IVIG in COVID-19 patients. High levels of C5a in sera and C5aR1 receptor in myeloid cells have been described in severe COVID-19 patients (*Nature*. 2020 Dec;588(7836):146-150. doi: 10.1038/s41586-020-2600-6). C5a is a potent activator of neutrophils and monocytes and high levels of C5a has been observed and associated with the pathology of several disease such as sepsis, rheumatoid arthritis, inflammatory bowel disease, ischemia reperfusion injury, antiphospholipid syndrome, systemic lupus erythematosus, psoriasis and neurodegenerative disease among many others (Manthey et al. *The International Journal of Biochemistry & Cell Biology* 41 (2009) 2114–2117). Interestingly, Carvelli J et al. (*Nature*. 2020 Dec;588(7836):146-150. doi: 10.1038/s41586-020-2600-6) showed that C5a enhanced the production of the proinflammatory cytokines IL-6, TNF and CCL2, all three showing a decreasing trend in IVIG treated individuals. Moreover, anti-C5aR1 therapeutic monoclonal antibodies prevented the C5a-mediated recruitment and activation of human myeloid cells, and inhibited acute lung injury in a human C5aR1 knock-in mouse model, pointing out the importance of the C5a-C5aR1 axes in COVID-19 development. In addition to the effect on C5a and proinflammatory cytokines, and according with the multifunctional properties of the IVIG, we have also observed a decreasing trend in the levels of Th2 cytokines and HGF, which have been associated with severe COVID-19, immunosuppression and fatal outcome. Interestingly, a correlation among the levels of HGF, IL-10 and IL-16 has been observed (see response to reviewer 1).

The reviewer has paid attention to our hypothesis about the role of IVIG regulating the LPS signaling. We would like to emphasize that this is just a hypothesis about the putative effect of the gut leakage on the development of severe COVID-19. This hypothesis is also supported by the levels of FABP2. Moreover, a plasmatic increase in the levels of endotoxin, (1 \rightarrow 3)- β -d-glucan and bacterial DNA (markers of gut leakage) have also been described in severe COVID-19 patients (Sirivongrangson et al., 2020). About 12% of COVID-19 patients manifest gastrointestinal symptoms whereas SARS-CoV-2 shedding in feces is observed in 40% of infected individuals (Parasa et al., 2020). Although our results are purely speculative, it is important to emphasize that obese and diabetic patients show an increment in the levels of LPS in plasma (*Metabolism*, 2017 Mar;68:133-144; *Diabetes Care* 36:3627–3632, 2013) and these conditions are among the main comorbidities associated with severe COVID-19. Moreover, it has been reported that the Spike glycoprotein of SARS-CoV-2 can bind to LPS and promote inflammation in vitro and in vivo (Petruk et al., 2020). Therefore, although it is just an hypothesis, it is possible that LPS could contribute to the development of severe COVID-19 directly or indirectly, as it has been recently exposed by Kruglikov and Scherer (*PLoS Pathog* 17(2): e1009306).

We are aware about the controversies around hydroxychloroquine and many other drugs used for treating SARS-COV-2 infected individuals (anti-GM-CSF, anti-HIV drugs, azithromycin, etc). That is the reason why we want to be very careful with our interpretation. Actually, we have highlighted the main limitations of our study and, at the end of the manuscript, we have stated that the observations made in the manuscript need to be confirmed in larger clinical trials. However, we would also like to highlight and point out to the case of tocilizumab (anti-IL6R antibody) (Gupta S and Leaf DE. Lancet. 2021 May 1;397(10285):1599-1601.doi: 10.1016/S0140-6736(21)00712-1). Early observations based on small studies suggested that tocilizumab could improve the outcome of hospitalized COVID-19 patients. Posterior clinical trials showed no benefit (N Engl J Med 2021;384:1503-16. DOI: 10.1056/NEJMoa2028700). However, the RECOVERY study showed that tocilizumab improved the outcome of hospitalized COVID-19 patients with hypoxia and systemic inflammation (Lancet 2021; 397: 1637–45). For us, all these controversies highlight the need to conduct proper clinical trials to confirm early data generated using small cohorts or pilot studies but, at the same time, reinforce the importance of paying attention to this early studies that can be key to identify signals that could translate into direct benefit and wellbeing of hospitalized patients.

Finally, we agree with the reviewer that some relevant references needed to be incorporated in the manuscript. We have included them in the revised version.

July 1, 2021

RE: Life Science Alliance Manuscript #LSA-2020-01009R

Dr. Jorge Carrillo
IrsiCaixa Institute for AIDS Research
Ctra del Canyet s/n
Hospital Universitario Germans Trias i Pujol
Badalona, Barcelona 08912
Spain

Dear Dr. Carrillo,

Thank you for submitting your revised manuscript entitled "High-dose intravenous immunoglobulins might modulate inflammation in COVID-19 patients.". We would be happy to publish your paper in Life Science Alliance pending final revisions necessary to meet our formatting guidelines.

- please add ORCID ID for secondary corresponding author-she should have received instructions on how to do so
- please separate the Results and Discussion section into two - 1. Results 2. Discussion, as per our formatting requirements
- please use the [10 author names, et al.] format in your references (i.e. limit the author names to the first 10)
- please add your main, supplementary figure, and table legends to the main manuscript text after the references section
- please add a callout for Figure S2B to your main manuscript text
- please add a link in the data availability section so that readers could easily access to the full deposited data set without having to contact the corresponding author
- please include an approval statement for the human samples used in this study

A. FINAL FILES:

B. MANUSCRIPT ORGANIZATION AND FORMATTING:

Sincerely,

Eric Sawey, PhD
Executive Editor
Life Science Alliance

<http://www.lsajournal.org>

Reviewer #1 (Comments to the Authors (Required)):

The authors investigate and describe changes of selected soluble factors in response to MIG treatment in COVID-19, this presents an important input in future studies addressing both effects of MIG in COVID-19 and possibly other hyper-inflammatory conditions. The control group from already published data strengthen the results and hypotheses generated. The reviewer has no further concerns and thanks the authors for their thorough revision of the paper.

Reviewer #2 (Comments to the Authors (Required)):

The authors managed to resubmit an improved version of their manuscript. the manuscript is much clearer and figures are presenting proper statistics. The authors managed to display control group information which considerably strengthen the results observed in MIG group. The response to reviewers highlight their hypothesis too. Based on this manuscript i recommend the editor to accept this article for publication.

July 12, 2021

RE: Life Science Alliance Manuscript #LSA-2020-01009RR

Dr. Jorge Carrillo
IrsiCaixa Institute for AIDS Research
Ctra del Canyet s/n
Hospital Universitario Germans Trias i Pujol
Badalona, Barcelona 08912
Spain

Dear Dr. Carrillo,

Thank you for submitting your Research Article entitled "High-dose intravenous immunoglobulins might modulate inflammation in COVID-19 patients". It is a pleasure to let you know that your manuscript is now accepted for publication in Life Science Alliance. Congratulations on this interesting work.

DISTRIBUTION OF MATERIALS:

Again, congratulations on a very nice paper. I hope you found the review process to be constructive and are pleased with how the manuscript was handled editorially. We look forward to future exciting submissions from your lab.

Sincerely,
